# The Communication Satisfaction of Geriatric Patients Treated by Dental Students and Dentists in a University Dental Clinic: A Cross-Sectional Study

**DOI:** 10.3390/geriatrics10040093

**Published:** 2025-07-14

**Authors:** Carla Semedo, Joana Costa, Elisa Kern de Castro

**Affiliations:** 1Master’s Degree in Dental Medicine, Egas Moniz School of Health & Science, 2829-511 Almada, Portugal; 2Clinical Research Unit (CRU), Egas Moniz Center for Interdisciplinary Research (CiiEM), Egas Moniz School of Health & Science, 2829-511 Almada, Portugal; jcosta@egasmoniz.edu.pt

**Keywords:** communication, patient-centered care, geriatrics, aging, patient satisfaction

## Abstract

**Background/Objectives**: With aging linked to increased oral health conditions, the communication skills of dental professionals are vital to ensure patient satisfaction and improve the quality of geriatric dental care. This cross-sectional study evaluated geriatric patients’ satisfaction with communication at a university dental clinic, comparing interactions with dentists and fifth-year dental students. **Methods**: A self-report questionnaire was administered to 111 patients, assessing sociodemographic data, general health, and satisfaction across six communication dimensions: verbal and nonverbal communication, empathy, respect, problem-solving, and support materials. The data were analyzed using Mann–Whitney U tests (α = 0.05). **Results**: The geriatric patients exhibited higher levels of satisfaction when interacting with dentists compared to students, particularly in the domains of nonverbal communication (*p* = 0.007), empathy (*p* = 0.035), and respect (*p* = 0.017). However, no statistically significant differences (*p* > 0.05) were observed in terms of verbal communication, problem-solving, and support materials. **Conclusions**: The geriatric patients demonstrated greater satisfaction with interactions with practicing dentists. These findings indicate that an attending dentist’s clinical experience enhances interpersonal interactions with geriatric patients. Therefore, developing the interpersonal skills of future dentistry professionals, as well as adapting communication to the needs of the elderly, is essential to provide more satisfactory experiences in geriatric dental care.

## 1. Introduction

The increase in the elderly population is a global phenomenon with significant impacts on public health, especially in European countries such as Portugal [1]. In Portugal, life expectancy has risen substantially in recent decades, reaching an estimated 80.96 years during the 2020–2022 period (78.05 years for males and 83.52 years for females) [2]. According to Eurostat (the Statistical Office of the European Union), Portugal is projected to have one of the highest proportions of elderly individuals in the European Union [3].

Aging is associated with an increased prevalence of chronic and disabling diseases, including cardiovascular diseases, diabetes, dementia, and oral health conditions [4].

Effective communication can be defined as the process by which a message is transmitted clearly and precisely, ensuring its accurate reception by the intended recipient [5]. In dental care, it strengthens the dentist–patient relationship and improves the overall quality of treatment. Dentists who communicate clearly and empathetically increase patients’ trust and confidence, particularly in managing emotional responses during procedures. This communication also reduces dental anxiety, which often leads to avoidance of care and worsening oral health [6]. In the context of geriatric patients, effective communication is particularly crucial due to the high prevalence of oral pathologies, such as periodontal disease, dental caries, xerostomia, and oral mucosal lesions. These conditions often result from systemic diseases; frequent medication use; and other coexisting factors, including alcohol and tobacco consumption, as well as the use of dental prostheses. Communication is one of several functional domains, including cognition, mobility, and mood, that significantly influence the health and well-being of elderly patients [7].

Effective communication has been shown to provide numerous benefits for elderly patients, including facilitating disease diagnosis, enhancing patient understanding and adherence to treatment regimens, reducing stress for both patients and healthcare professionals, and minimizing unnecessary healthcare expenditures (e.g., redundant tests and consultations) [8]. In contrast, communication failures can deteriorate the patient–provider relationship, leading to difficulties in establishing diagnoses, reduced patient comprehension of their condition, and increases in patient dissatisfaction and complaints [6,8,9,10].

Traditionally, dental education has its origins in the biomedical model [11], which mainly emphasizes technical skills and scientific knowledge, focusing on the biological aspects of dental care [12]. However, there is a growing recommendation among experts that dental training and practice should adopt a more patient-centered approach [6,11,13,14,15]. This model prioritizes personalized care that addresses patients’ individual needs and concerns, actively involving them in decisions about their health [14]. 

Patient satisfaction is a multifaceted concept that reflects individuals’ subjective assessments of the quality of healthcare services they receive [16]. It refers to the degree of contentment or pleasure experienced by patients when utilizing a health service [17]. Satisfaction is closely linked to people’s expectations of the services received, based on health, illness, quality of life, and other needs [18]. Studies have highlighted the importance of trust, active listening, and positive interpersonal relationships in achieving patient satisfaction, particularly among elderly populations. Communication skills and active listening have been identified as critical factors with major positive impacts on patient satisfaction. Additionally, other factors—such as the quality of care, the accessibility of services, the physical environments of healthcare facilities, and treatment effectiveness—have been extensively documented in the literature.

The main purpose of this research was to assess whether there are differences in the communication satisfaction of geriatric patients treated at a university dental clinic when interacting with dentists compared to fifth-year undergraduate dental students.

## 2. Materials and Methods

### 2.1. Design and Ethical Procedures

This cross-sectional and exploratory study was conducted at a university dental clinic (Egas Moniz Dental Clinic, Almada, Portugal) between March and May 2024. This study was approved by the Ethics Committee of Egas Moniz School of Health & Science on 27 February 2024 (approval number: PT-310/23) and followed the rules of the Declaration of Helsinki of 1975, which were revised in 2013. All participants signed an informed consent participation agreement.

### 2.2. Sample and Data Collection Procedures

The participants were invited to participate in this study after their dental appointments. All participants were informed about the research objectives and the anonymous and voluntary nature of their participation. The inclusion criteria selected geriatric patients (aged 65 years or older) who were fluent in Portuguese and were attended by dentists or fifth-year (final-year) dental students. The exclusion criteria removed patients with self-reported psychiatric disorders and/or vision, hearing, or speech impairments. The time expended in responding to the questionnaire was between 15 and 20 min.

### 2.3. Measures

The following instruments were used for data collection:

Sociodemographic Questionnaire: This questionnaire collected information on age, biological sex, nationality, primary language, marital status, number of children, living arrangements, geographical location, educational qualifications, work regime, and the frequency of visits to the institution’s dental clinic within the previous 12 months.

General Health Status Questionnaire: This self-report questionnaire included the reason for the consultation; diagnosis of chronic or cognitive illness; and the presence of hearing, speech, or vision problems, in order to confirm that the participants met the eligibility criteria and should not have been excluded from the study.

Communication Satisfaction Questionnaire [19]: This instrument evaluated patient satisfaction in communication with healthcare professionals based on 31 items divided into six subscales: verbal communication (6 items), nonverbal communication (4 items), empathy (4 items), respect (4 items), problem-solving (3 items), and support material (4 items). Each dimension was assessed using a five-point Likert scale ranging from 1 (very dissatisfied) to 5 (very satisfied) (1 = very dissatisfied; 2 = dissatisfied; 3 = neither satisfied nor dissatisfied; 4 = satisfied; 5 = very satisfied; N/A = not applicable). In the “verbal communication” dimension, the inquiries encompassed communication in the form of speech, as how healthcare professionals interact with patients is of paramount importance. Verbal and nonverbal communication between healthcare professionals and patients plays a crucial role in establishing a therapeutic rapport. The “nonverbal communication” dimension encompassed inquiries designed to assess the involvement, expressions, and emotions of the healthcare professional, thereby facilitating patient comprehension of verbal communication. The “empathy” dimension involved the utilization of specific inquiries aimed at gauging the healthcare professional’s capacity to empathetically understand the patient’s emotional state. The “respect” dimension encompassed questions that evaluated the ethicality and respectfulness of the healthcare professional’s approach towards the patient. The “problem-solving” dimension encompassed inquiries into the extent to which healthcare professionals possess the capacity to manage unanticipated occurrences that may transpire and whether they are equipped to apprise patients of forthcoming events. The “support material” dimension pertained to the resources furnished by healthcare professionals to facilitate patient–professional interactions. As the nomenclature of this dimension suggests, support materials that can assist with treatment and related processes. The Portuguese-language questionnaire is available as Appendix A.

### 2.4. Data Analysis

Statistical analysis was carried out using the IBM Statistical Package for the Social Sciences (SPSS) statistics program (version 27.0.1.0 for Windows). Descriptive statistics were calculated for the results in general. The Shapiro–Wilk test (*p* < 0.05) indicated that the data did not follow a normal distribution. Given the non-parametric nature of the data, Mann–Whitney U tests were used to compare the levels of satisfaction between the geriatric patients treated by dentists and those treated by fifth-year undergraduate students across the different dimensions of communication. A significance level of 0.05 was chosen to determine statistical significance before the analysis.

## 3. Results

### 3.1. The Sociodemographic and General Health Characteristics of the Sample

The study sample consisted of 111 geriatric patients, with a mean age of 71.78 years (±5.11), ranging from 65 to 87 years old. Only three patients were excluded because of vision or hearing problems.

Table 1 presents the sociodemographic characteristics of the geriatric patients treated by dentists and undergraduate dental students.

The sample exhibited a balanced distribution based on biological sex, with a preponderance of Portuguese patients in both groups, underscoring cultural and linguistic homogeneity. Most patients in both groups were married or cohabiting, had children, and resided in urban areas. Educational attainment varied significantly, and most of the participants had not completed the 12th grade. Additionally, the majority of the patients in both groups were retired. Concerning the prevalence of self-reported chronic diseases, 38.7% of the total sample (45.6% of the patients examined by the students and 31.5% of the patients examined by the dentists) reported receiving a diagnosis of a chronic disease.

### 3.2. Satisfaction with the Communication

Table 2 summarizes the means, standard deviations, interquartile ranges (IQRs), minima, and maxima for the six dimensions of satisfaction: verbal communication, nonverbal communication, empathy, respect, problem-solving, and support materials received.

The results of communication satisfaction, as shown in Table 3, revealed significant differences between the patients treated by dentists and undergraduate dental students across several dimensions. The geriatric patients treated by dentists reported significantly greater satisfaction with nonverbal communication than those treated by dental students. Similarly, the patients treated by dentists exhibited higher satisfaction with empathy in communication (*p* = 0.035) compared to those seen by undergraduates. A similar pattern was observed for satisfaction with respect in communication, with the patients seen by dentists perceiving a higher level of respect (*p* = 0.017) in communication compared to those seen by undergraduates. In contrast, no statistically significant differences (*p* > 0.05) were observed between the two groups in terms of verbal communication, problem-solving, and the use of support materials.

## 4. Discussion

Europe, particularly Portugal, is experiencing a significant increase in its elderly population. The present study provides relevant insights into communication with patients from this growing demographic segment.

The objective of this study was to evaluate the satisfaction of geriatric patients at a university clinic regarding communication with fifth-year undergraduate dental students and dentists. This study identified areas where geriatric patients perceive communication as satisfactory and areas needing improvement in dental education.

The findings indicated disparities in communication satisfaction among elderly patients, who expressed greater satisfaction with dentists compared to students. These disparities were particularly evident in the domains of nonverbal communication, empathy, and respect. One potential explanation for this discrepancy is that dentists’ extensive clinical experience plays a role in the development of these non-technical abilities. This assertion is further substantiated by previous studies, which have demonstrated that years of clinical practice are positively associated with the enhancement of communication skills among healthcare professionals [8,20,21]. Similarly, another study examined the communication skills of medical students before and after clinical internships, finding that the participants demonstrated significant improvements in areas such as empathy and information-gathering skills as they gained more practical experience [22]. The findings of the present study suggest that experience with elderly patients accumulated during clinical practice may have enhanced the dentists’ communication skills, as reflected in patient satisfaction. These results also highlight an opportunity to improve the interpersonal skills of dental students, particularly in PROOthe areas/domains of nonverbal communication, empathy, and respect, through targeted training.

One factor contributing to the challenges faced by dental students in interacting with patients is the passive nature of their communication skills development in higher education. Academic instructors may encounter difficulties in evaluating their students’ abilities, leading to limited implementation of active teaching methodologies that integrate verbal and nonverbal communication [9]. Furthermore, the communication skills of undergraduate students can be influenced by various factors, including stress associated with clinical procedures, assessment pressures, limited leisure time, fatigue, and emotional exhaustion [21]. In this sense, it is important to have a structured teaching approach for the professional training of dental students integrated into their curriculum, which can include role-playing and interviews with patients to train active listening skills, empathy, and information sharing, among other aspects of patient-centered communication [6].

Regarding the remaining subscales, specifically verbal communication, problem-solving, and support materials, the findings indicated that there were no differences in the satisfaction levels of the elderly patients when interacting with dentists or undergraduates. These results suggest that the students demonstrated proficiency in these communication domains. It has been demonstrated that undergraduates, at the culmination of their program, exhibit notable self-assurance in their technical dentistry knowledge [23], which can contribute to the enhancement of their communication skills [24]. This progression is evident in the capacity to more effectively articulate diagnoses, treatment strategies, and the management of any clinical challenges.

Our findings align with those of Kee et al. [8] regarding nonverbal communication, empathy, and respect. Their study identified issues such as inadequate attitudes, lack of eye contact, negative body posture, and a paucity of empathy among newly graduated professionals. These deficiencies can be attributed to the fact that new graduates are still developing their communication skills. Furthermore, the researchers attributed the observed challenges, including the lack of eye contact and negative body posture, to the prevalent use of computers in office environments.

To develop effective communication skills, students must receive training accompanied by constant coaching. The communication training curriculum should prioritize patient-centered care, active listening, role-playing, video recording, and practical experience with actual patients [25]. Integration of scientific knowledge and interpersonal skills into dental courses is recommended, with educational videos and simulations of interactions between students and patients being particularly useful. Early exposure to patients has been identified as a key strategy to enhance students’ communication skills, fostering a more realistic and empathetic understanding of their future professional encounters [26].

The reasons for the patients’ dental appointments were not recorded, which limits the ability to analyze whether communication satisfaction varies according to treatment type or complexity. Different clinical procedures may require different communication approaches, and this should be considered in future studies.

This study is inherently limited in its scope and generalizability. As it used a convenience sample limited to a single university clinic, the results are not representative of the population and cannot be generalized. Moreover, individuals with speech and other communication impairments were excluded, which represents another limitation. Future studies should consider adapted methodologies to include these important patient groups and ensure a more inclusive understanding of communication challenges in dental care. Additionally, while we did not assess socioeconomic status directly, the available data suggest that the majority of our participants likely belong to lower socioeconomic groups, which may also influence the findings and limit broader applicability. Nevertheless, the findings offer valuable indications regarding areas for improvement in communication with elderly patients in dental practice. Additionally, during data collection, some patients expressed dissatisfaction with the length of the questionnaire, which may have led to fatigue.

This was a non-blinded study, as the participants were aware of whether they were being treated by dental students or practicing dentists. This awareness may have introduced response bias, potentially influencing how the patients rated their satisfaction with communication.

It is suggested that future studies replicate and expand the number of participants, including patients seen by students from different years of the dental course, to allow a broader analysis of their skills. Longitudinal studies that systematically track changes in patient satisfaction resulting from communication training over an extended period would be ideal. Furthermore, incorporation of objective data derived from behavioral observations during dental consultations could serve to mitigate the potential for social desirability bias, which is often present in self-report questionnaires.

## 5. Conclusions

Elderly patients report a higher level of satisfaction when communicating with dentists compared to undergraduates, particularly in specific areas that can be improved, such as nonverbal communication, empathy, and respect. Thus, implementing targeted educational strategies to enhance students’ communication skills is essential.

To address the identified gaps, a robust educational approach is necessary to develop students’ interpersonal skills, especially in nonverbal communication and empathy, which are fundamental to the satisfaction of elderly patients. Strategies such as role-playing, educational videos, and simulations, as well as increased practical exposure to patients beginning in the early stages of the dental program, have been shown to be effective in enhancing these competencies. Implementing such measures could significantly improve the quality of communication between dental students and elderly patients, ultimately fostering better patient outcomes.

## Figures and Tables

**Table 1 geriatrics-10-00093-t001:** Descriptive statistics of the sociodemographic characteristics of the participants.

Characteristics		Total Sample (*n* = 111)	Patients Treated by Dentists (*n* = 54)	Patients Treated by Undergraduate Students (*n* = 57)
Biological sex	Female	53 (47.7%)	25 (46.3%)	28 (49.1%)
	Male	58 (52.3%)	29 (53.7%)	29 (50.9%)
Nationality	Portuguese	108 (97.3%)	53 (98.1%)	55 (96.5%)
	Others	3 (2.7%)	1 (1.9%)	2 (3.6%)
Portuguese primary language	Yes	108 (97.3%)	53 (98.1%)	55 (96.5%)
	No	3 (2.7%)	1 (1.9%)	2 (3.5%)
Marital status	Single	15 (13.5%)	5 (9.3%)	10 (17.5%)
	Married	62 (55.9%)	30 (55.6%)	32 (56.1%)
	Divorced	12 (10.8%)	7 (13.0%)	5 (8.8%)
	Widowed	22 (19.8%)	12 (22.2%)	10 (17.5%)
Children	Yes	86 (77.5%)	41 (75.9%)	45 (78.9%)
	No	25 (22.5%)	13 (24.1%)	12 (21.1%)
Living alone	Yes	36 (32.4%)	19 (35.2%)	17 (29.8%)
	No	75 (67.6%)	35 (64.8%)	40 (70.2%)
Live in an urban area	Yes	107 (96.4%)	54 (100%)	53 (93.0%)
	No	4 (3.6%)	0 (0%)	4 (7.0%)
Education level	Up to 4th grade	38 (83.4%)	15 (27.8%)	23 (40.4%)
	From 5th to 9th grade	29 (26.1%)	13 (24.1%)	16 (28.1%)
	From 9th to 12th grade	27 (24.3%)	18 (33.3%)	9 (15.8%)
	University	17 (15.3%)	8 (14.8%)	9 (15.9%)
Employment status	Full-time	8 (7.2%)	3 (5.6%)	5 (8.8%)
	Retired	101 (91.0%)	50 (92.6%)	51 (89.5%)
	Unemployed	2 (1.8%)	1 (1.9%)	1 (1.8%)
The frequency of visits to the institution’s dental clinic within the previous 12 months	1	5 (4.5%)	2 (3.7%)	3 (5.3%)
	2	13 (11.7%)	10 (18.5%)	3 (5.3%)
	3 or more	93 (83.8%)	42 (77.8%)	51 (89.5%)
Chronic disease	yes	43 (38.7%)	17 (31.5%)	26 (45.6%)

*n*—number of participants.

**Table 2 geriatrics-10-00093-t002:** Descriptive statistics of the satisfaction with the communication measures.

Variables	Mean (SD)	Median	IR	Min	Max
Verbal communication	4.77 (0.52)	5.00	5.00–5.00	3.00	5.00
Nonverbal communication	4.72 (0.68)	5.00	5.00–5.00	1.00	5.00
Empathy	4.76 (0.43)	5.00	5.00–5.00	2.00	5.00
Respect	4.85 (0.43)	5.00	5.00–5.00	2.00	5.00
Problem-solving	4.41 (0.69)	5.00	4.00–4.00	2.00	5.00
Support materials received	4.60 (0.62)	5.00	5.00–4.00	2.00	5.00

SD—standard deviation; IR—interquartile range; Min—minimum; Max—maximum.

**Table 3 geriatrics-10-00093-t003:** Comparison of satisfaction with communication between patients treated by dentists and those treated by undergraduate students.

Variables	Dentists Mean Rank	Undergraduate Students Mean Rank	U	*p*
Verbal communication	58.94	53.21	1380.000	0.160
Nonverbal communication	61.92	50.39	1219.500	0.007
Empathy	60.50	51.74	1296.000	0.035
Respect	60.44	51.79	1299.000	0.017
Problem-solving	55.98	56.02	1538.000	0.995
Support material received	55.03	56.92	1486.000	0.708

U: Mann–Whitney non-parametric U test.

## Data Availability

The data presented in the current study are available on request from the corresponding author.

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
