# Peer review of "The Communication Satisfaction of Geriatric Patients Treated by Dental Students and Dentists in a University Dental Clinic: A Cross-Sectional Study"

_geriatrics, 2025, doi:10.3390/geriatrics10040093_

Round 1
Reviewer 1 Report
Comments and Suggestions for Authors
Thank you very much for the interesting paper „Communication satisfaction of geriatric patients attending by dental students and dentists in a university dental clinic“
To improve the quality of the paper, I suggest to add the questionnaire or parts of it for better understanding what kind of questions were asked to evaluate the nonverbal pattern.
There is only mentioned: nonverbal communication, empathy and respect, but no clear measures.
In table 1 there is a shift in the colum „education level“, please correct.
As this study was a non blinded study, this should also be considered in the discussion.
Author Response
Comments 1: Thank you very much for the interesting paper „Communication satisfaction of geriatric patients attending by dental students and dentists in a university dental clinic“. To improve the quality of the paper, I suggest to add the questionnaire or parts of it for better understanding what kind of questions were asked to evaluate the nonverbal pattern. |
Response 1: Thank you for your valuable feedback and positive assessment of our manuscript. As suggested, we have included the full questionnaire as a supplementary file to provide a clearer understanding of the questions used, particularly those related to the evaluation of nonverbal communication patterns.
|
Comments 2: There is only mentioned: nonverbal communication, empathy and respect, but no clear measures. |
Response 2: Thank you for your observation. We agree that more clarity regarding the measurement of nonverbal communication, empathy, and respect is important. To address this, we have included the full Communication Satisfaction Questionnaire as a supplementary file. This questionnaire contains the specific items used to assess each of the six dimensions, including nonverbal communication, empathy, and respect. We have also clarified the structure and number of items per dimension in the Methods section to improve transparency.
Comments 3: In table 1 there is a shift in the colum „education level“, please correct. Response 3: Corrected.
Comments 4: As this study was a non blinded study, this should also be considered in the discussion. Response 4: Thank you for your important observation. We acknowledge that the non-blinded nature of the study may have introduced potential bias in participants’ responses. We have now addressed this limitation in the Discussion section, noting that the awareness of being evaluated by students or dentists could have influenced patients’ satisfaction ratings: “This was a non-blinded study, as participants were aware of whether they were being treated by dental students or attending dentists. This awareness may have introduced response bias, potentially influencing how patients rated their satisfaction with communication”.
|

Reviewer 2 Report
Comments and Suggestions for Authors
Study design must to be provided in the title and in the abstract.
Reason for appointments should be provided. I think is important to evaluate type of communication also basing on the type of treatments.
Conclusions must to be moved in the discussion. Conclusions must only to reflect the conclusions of the results (questionnaire).
Author Response
Comments 1: Study design must to be provided in the title and in the abstract. |
Response 1: Thank you for your suggestion. We have updated the manuscript title and the abstract to explicitly mention the study design, clarifying that this was a cross-sectional study.
|
Comments 2: Reason for appointments should be provided. I think is important to evaluate type of communication also basing on the type of treatments. |
Response 2: Thank you for your insightful comment. We agree that the type of treatment could influence communication and patient satisfaction. However, in this study, the specific reason for each appointment was not recorded or analyzed. We have now acknowledged this point as a limitation in the Discussion section and identified it as an important aspect to consider in future research (page 6, lines 221–224).
Comments 3: Conclusions must to be moved in the discussion. Conclusions must only to reflect the conclusions of the results (questionnaire). Response 3: Thank you for your thoughtful comment. We agree that the Conclusions section should focus strictly on the findings derived from the questionnaire. Following your suggestion, we have revised the manuscript by moving three paragraphs (page 6, lines 225–237) — which provided broader interpretation and contextual reflection — from the Conclusions to the Discussion section. The Conclusions section now strictly addresses the study’s objectives and summarizes the key results. This adjustment has improved the structure and clarity of the manuscript.
|

Reviewer 3 Report
Comments and Suggestions for Authors
Dear Authors
The topic of the manuscript is interesting, but some changes are necessary before considering it for publication. Here are my suggestions to improve it:
- The exclusion criteria include patients with vision, hearing, or speech impairment. Please specify what kind of vision impairment you refer to. Completely blind? Or the bare minimum to be able to read a questionnaire? And if it were the latter case, why not provide a questionnaire written in big characters, knowing full well that it is aimed at an elderly person with visual difficulties? Furthermore, did you consider having a person to help the elderly person with visual difficulties to fill out a questionnaire, considering that they should read and fill out an anamnesis and informed consent?
2 The same goes for patients with speech impairments... moreover, they also evaluate the doctors' non-verbal communication... who better than someone who can't hear or speak can evaluate this parameter? Furthermore, if you decide to use them as exclusion criteria, you should at least argue a little about the reasons, also because these patients are often accompanied by caregivers who can help in filling out the questionnaires
3 The socio-demographic questionnaire does not include the economic level at which the elderly person lives. Please specify.
4. Better specify the items in which the subscales were divided.
5 In Table 1, the "characteristics" item is not centered, and there is an n% that makes no sense to be put there. Please correct.
6 There is no flow chart showing the pre-questionnaire work, how many patients were included, and how many were discarded by the exclusion criteria.
7 The questionnaire files are missing. It would be better to upload them.
8- 9/25 references are more than 10 years old. If not strictly necessary, please change them.
Please address these considerations.
Best regards
Author Response
- The exclusion criteria include patients with vision, hearing, or speech impairment. Please specify what kind of vision impairment you refer to. Completely blind? Or the bare minimum to be able to read a questionnaire? And if it were the latter case, why not provide a questionnaire written in big characters, knowing full well that it is aimed at an elderly person with visual difficulties? Furthermore, did you consider having a person to help the elderly person with visual difficulties to fill out a questionnaire, considering that they should read and fill out an anamnesis and informed consent?
We thank the reviewer for this important observation. The exclusion criterion related to vision impairment specifically refers to individuals with visual difficulties that would prevent them from reading and accurately answering the questionnaire.
While we recognize that adaptations such as larger print or assistance from a third party could potentially help some patients, we opted not to implement these alternatives for two main reasons. First, dental appointments are often a stressful experience for elderly patients, and we aimed to minimize any additional sources of anxiety. Second, the university dental clinic where the study was conducted has a high patient flow and limited space, leading to a busy and sometimes noisy environment. These conditions could further increase patients' stress levels and make it difficult for them to concentrate on the questionnaire, even with support.
Therefore, in the interest of maintaining a calm and minimally intrusive experience for participants, we included as an exclusion criterion those individuals whose visual impairment would significantly interfere with their ability to independently complete the questionnaire.
2 The same goes for patients with speech impairments... moreover, they also evaluate the doctors' non-verbal communication... who better than someone who can't hear or speak can evaluate this parameter? Furthermore, if you decide to use them as exclusion criteria, you should at least argue a little about the reasons, also because these patients are often accompanied by caregivers who can help in filling out the questionnaires
The exclusion of patients with speech impairments was based on practical and methodological considerations related to the design and objectives of our study.
While we agree that individuals with speech impairments may be particularly sensitive to non-verbal cues—and could, in theory, provide valuable insights—we were also concerned with ensuring that all participants could consistently and autonomously engage with the full range of questionnaire items.
Moreover, while we recognize that caregivers often accompany patients with communication challenges and could assist in completing the questionnaire, we chose to limit third-party involvement to preserve the subjective nature of the responses. Including caregiver-assisted answers could introduce variability in interpretation and potentially affect the reliability and comparability of the data across the sample.
We acknowledge the reviewer’s point and agree that future studies specifically designed to evaluate the experiences of patients with speech or hearing impairments, using adapted instruments and methodologies, could provide important and complementary perspectives on non-verbal communication in dental settings.
Considering the reviewer’s comment, we have included a discussion of this limitation in the manuscript, acknowledging that the exclusion of individuals with speech and other communication impairments restricts the comprehension of our findings and that future studies should consider adapted methodologies to include these important patient groups.
3 The socio-demographic questionnaire does not include the economic level at which the elderly person lives. Please specify.
We thank the reviewer for this important observation. It is correct that our socio-demographic questionnaire did not include a direct question regarding the participants’ socioeconomic status. We recognize this as a limitation in our data collection instrument and acknowledge it as an oversight.
However, we did collect related information that allows us to infer the general socioeconomic profile of our sample. Specifically, we gathered data on participants' educational level and previous or current profession. From this, we observed that 34.2% of participants had only completed 4 years of schooling (basic level), and an additional 29.1% had studied up to 9th grade. This indicates that more than half of our sample had relatively low levels of formal education, which often correlates with lower socioeconomic status.
Furthermore, it is important to note that the dental care provided at the university clinic is offered at significantly reduced costs, which generally attracts individuals from lower-income backgrounds. Therefore, while we did not assess socioeconomic status directly, the available data suggest that the majority of our participants likely belong to lower socioeconomic groups.
We have now addressed this limitation in the discussion section of the revised manuscript.
- Better specify the items in which the subscales were divided.
Thank you for your suggestion. The full original questionnaire, including the specific items comprising each subscale, is available in the doctoral dissertation by Santos, A. H. R. D. (2013), titled Avaliação da satisfação do utente na comunicação com os profissionais de saúde: construção e validação de um instrumento (Instituto Politécnico do Porto, Escola Superior de Tecnologia da Saúde do Porto, Instituto Politécnico de Bragança, Instituto Politécnico do Cávado e do Ave, Instituto Politécnico de Viana do Castelo). We believe that including a detailed description of all questionnaire items within the main text of the paper may be excessive and potentially tiresome for readers. However, we remain open to including a summary or additional details in a supplementary file if the reviewer or editors consider it necessary.
5 In Table 1, the "characteristics" item is not centered, and there is an n% that makes no sense to be put there. Please correct.
Thank you for your observation. The formatting issue in Table 1 has been corrected—the "Characteristics" header is now properly centered, and the unnecessary "n%" has been removed. Everything is now in order.
6 There is no flow chart showing the pre-questionnaire work, how many patients were included, and how many were discarded by the exclusion criteria.
We have clarified the participant selection process in the manuscript. As this was a convenience sample, we invited patients who were present in the waiting room of the dental clinic during the study period. A total of 114 patients were invited to participate. Of these, 3 were excluded based on the exclusion criteria, resulting in a final sample of 111 participants. This information has been added to the Methods section for clarity. We did not include a flow chart, as the process was straightforward and fully described in the text.
7 The questionnaire files are missing. It would be better to upload them.
Thank you for your observation. As mentioned in the manuscript, the full questionnaire is available in Santos (2013), which is the original source of the instrument. Since we are not the authors of the questionnaire, we cannot include it as an appendix in the manuscript. However, we are happy to upload the original version (in Portuguese) as supplementary material, with proper attribution to the original author.
8- 9/25 references are more than 10 years old. If not strictly necessary, please change them.
Please address these considerations.
We appreciate the opportunity to clarify their relevance. Many of the cited sources, although more than 10 years old, are foundational and remain highly relevant to the theoretical framework of patient-centered care—such as Apelian (2014), Bardes (2012), and Erikson et al. (2008). These works continue to be widely referenced and are essential for contextualizing our study.
Additionally, some of the older references are specific to our topic within dentistry. For example, Carey et al. (2010) provides important insights into communication skills in dental education, which remain pertinent to our discussion.
The reference by Santos (2013) is particularly crucial, as it is the original source for the instrument used in our study and, therefore, cannot be replaced.
However, in response to your suggestion, we have excluded the reference by Mindaye (2012) to reduce reliance on older literature.
We hope these adjustments address your concerns while preserving the integrity and relevance of our theoretical and methodological framework.

Reviewer 4 Report
Comments and Suggestions for Authors
Review Manuscript ID geriatrics-3546271
Dear Authors,
The topic of your research and the title of the manuscript are both highly interesting and relevant for improving the quality of dental care among the elderly population. However, I have identified a number of shortcomings that I kindly ask you to consider and address in your revised version, as outlined below:
Abstract:
The type of questionnaire used should be specified. The authors are requested to indicate in the abstract which statistical software was used for the analysis. Please check line 17, specifically the reference to the statistical significance threshold α = 0.05 and explain whether the threshold was set before the analysis.
Keywords:
It is recommended to include MeSH terms whenever possible.
Introduction:
All references in the text should be cited numerically and enclosed in square brackets. The alternation between numbered references (e.g., line 79, reference [16]) and in-text author name citations creates an impression of inconsistency and a lack of organization in the manuscript. The study's hypothesis is missing, as is the authors’ motivation for choosing this particular topic — both would be valuable to include. It should also be stated — either in the Introduction under the study's aim or in the Materials and Methods section — whether fifth-year dental students at the University of Almada are in their final year of study. Additionally, the quality of the problem statement in the Introduction should be improved. I recommend incorporating more recent literature, such as:
Ho, J.C.Y.; Chai, H.H.; Luo, B.W.; Lo, E.C.M.; Huang, M.Z.; Chu, C.H. An Overview of Dentist–Patient Communication in Quality Dental Care. Dent. J. 2025, 13, 31. https://doi.org/10.3390/dj13010031
Güler, Kübra, and Emine Pirim Görgün. "A cross-sectional survey study on the use of communication methods in the dentist-geriatric patient relationship." Cumhuriyet Dental Journal 25.3 (2022): 230–238.
Hamasaki, Tomoko, and Akihito Hagihara. "A study of dentists’ explanations and patient-dentist communication among older adults in Japan." BMC Oral Health 24.1 (2024): 1163.
Materials and Methods:
Under the exclusion criteria (line 97), you mentioned that subjects with speech, vision, or hearing impairments were excluded. At the same time, the General Health Status Questionnaire includes questions regarding the presence of hearing, vision, and speech problems (line 105), which creates a contradiction. Please clarify and resolve this inconsistency. Additionally, you need to state the initial sample size, especially since you mention exclusion criteria. Indicate please how many participants were excluded.
More detailed information about the questionnaire should be provided to clearly describe it. The questionnaire itself should either be included within the Materials and Methods section or attached as a supplementary file. The estimated time required to complete the questionnaires — specifically the three instruments applied to the convenience sample — should be stated.
Finally, it appears that many data points were collected but not used in your analysis, such as sociodemografic and health data. Please clarify the rationale for collecting these data and specify whether they were intended for future analyses, exploratory purposes, or were later deemed irrelevant.
Results:
The data presented on line 134 (38.7% self-reported chronic diseases), as well as the information from lines 135–139, do not appear in any of the tables. At the same time, sociodemographic data are included in Table 1, which reflects a disorganized presentation of the results.
Furthermore, both the sociodemographic variables and the self-reported general health data seem to have no relevance to your study, as they are not used or analyzed in any meaningful way.
I strongly recommend reorganizing the results section — either by grouping participants into age categories or by conducting correlational analyses to identify potentially vulnerable subgroups in terms of communication with the dentist or dental student. Otherwise, it remains merely a descriptive presentation of unused data, which does not contribute meaningfully to the study's objectives or conclusions.
You need to be consistent in how you report p-values — for example, line 151 reports p = .007, while line 156 uses p = 0.007. Please choose one format and apply it uniformly throughout the manuscript.
Discussion:
On line 116, reference number [10] appears to be incorrectly formatted or inconsistent with the citation style used throughout the manuscript — please revise accordingly.
You are encouraged to add recent comparative studies and to expand the discussion through relevant comparisons with similar findings in the current literature.
Moreover, the limitations of the study are missing and should be clearly stated.
Lastly, please provide more insights into how dental academic programs can implement active teaching methodologies that effectively integrate both verbal and non-verbal communication strategies into the curriculum.
Conclusions:
Please move the study limitations to the end of the Discussion section, where they more appropriately belong (e.g., “research conducted at a single dental university – limited external validity”).
References:
I kindly ask you to expand the reference list by including more recent and relevant sources. Additionally, the entire bibliography must be reformatted to match the citation style required by the target journal, as it currently does not conform to the submission guidelines.

Author Response
Abstract The type of questionnaire used should be specified. The authors are requested to indicate in the abstract which statistical software was used for the analysis. Please check line 17, specifically the reference to the statistical significance threshold α = 0.05 and explain whether the threshold was set before the analysis. Thank you for your comment. We have indicated in the abstract that a self-report questionnaire was used. Regarding the statistical significance threshold (α = 0.05) mentioned in line 17, we confirm that this threshold was defined prior to the analysis. The α = 0.05 level is commonly used in the social sciences and is the default significance level in SPSS analyses. We used SPSS version 27.0 for all statistical analyses. Although this information, along with the explanation of the significance threshold, is already included in the main text, we did not include it in the abstract due to space limitations. We hope this clarifies your concerns.
Keywords: It is recommended to include MeSH terms whenever possible. Thank you for the recommendation. We confirm that the terms "communication," "geriatrics," and "aging" are official MeSH terms. To ensure consistency with MeSH terminology, we have revised the keywords as follows:
Introduction: All references in the text should be cited numerically and enclosed in square brackets. The alternation between numbered references (e.g., line 79, reference [16]) and in-text author name citations creates an impression of inconsistency and a lack of organization in the manuscript. The study's hypothesis is missing, as is the authors’ motivation for choosing this particular topic — both would be valuable to include. It should also be stated — either in the Introduction under the study's aim or in the Materials and Methods section — whether fifth-year dental students at the University of Almada are in their final year of study. Additionally, the quality of the problem statement in the Introduction should be improved. I recommend incorporating more recent literature, such as: Ho, J.C.Y.; Chai, H.H.; Luo, B.W.; Lo, E.C.M.; Huang, M.Z.; Chu, C.H. An Overview of Dentist–Patient Communication in Quality Dental Care. Dent. J. 2025, 13, 31. https://doi.org/10.3390/dj13010031 Güler, Kübra, and Emine Pirim Görgün. "A cross-sectional survey study on the use of communication methods in the dentist-geriatric patient relationship." Cumhuriyet Dental Journal 25.3 (2022): 230–238. Hamasaki, Tomoko, and Akihito Hagihara. "A study of dentists’ explanations and patient-dentist communication among older adults in Japan." BMC Oral Health 24.1 (2024): 1163.
We have carefully reviewed the formatting of all references in the manuscript and apologize for the previous inconsistency. All references are now cited numerically and enclosed in square brackets, as per the journal’s guidelines, to ensure clarity and consistency throughout the text. Regarding the absence of a stated hypothesis, we would like to clarify that this study is exploratory. The limited availability of prior research on this specific topic, particularly within the Portuguese context, did not provide a clear foundation for developing a precise hypothesis. Furthermore, this is, to our knowledge, the first study addressing this topic in Portugal. We recognize that cultural and social characteristics can significantly influence both verbal and non-verbal communication, which further supports our decision to adopt an exploratory approach. We also confirm that the fifth-year dental students at the University of Almada are in their final year of study. This information has now been explicitly stated in the manuscript to provide appropriate context for the reader. Finally, we have read the recommended papers with great interest and have included the study by Ho et al. (2025) in our revised manuscript.
Materials and Methods: Under the exclusion criteria (line 97), you mentioned that subjects with speech, vision, or hearing impairments were excluded. At the same time, the General Health Status Questionnaire includes questions regarding the presence of hearing, vision, and speech problems (line 105), which creates a contradiction. Please clarify and resolve this inconsistency. When inviting patients to participate in the study, we verbally explained the research, read the informed consent aloud, and clearly outlined the exclusion criteria, including any speech, vision, or hearing impairments. Only individuals who met all eligibility requirements and agreed to participate were included in the study. The General Health Status Questionnaire was then administered to collect comprehensive health information and to confirm the eligibility of participants. Including questions about hearing, vision, and speech allowed us to validate that the criteria were appropriately applied and that participants did not present any of the conditions listed for exclusion. To avoid any potential confusion, we have revised the manuscript to more clearly explain this aspect of our methodology.
Additionally, you need to state the initial sample size, especially since you mention exclusion criteria. Indicate please how many participants were excluded. We have now included the initial sample size in the manuscript, as well as the number of participants excluded based on the defined criteria. Specifically, three patients were excluded due to the presence of speech, vision, or hearing impairments, in accordance with the exclusion criteria. This clarification has been added to ensure transparency in our sampling process.
More detailed information about the questionnaire should be provided to clearly describe it. The questionnaire itself should either be included within the Materials and Methods section or attached as a supplementary file. The estimated time required to complete the questionnaires — specifically the three instruments applied to the convenience sample — should be stated. We have added more detailed information about the self-report questionnaire in the Materials and Methods section to enhance clarity. However, we are unable to include the full instrument in the manuscript, as it is a previously developed and published measure. The full version of the questionnaire can be found in Santos (2013), the original source in which the instrument was created and validated. We have cited this reference appropriately in the revised text. Additionally, we have included the estimated time required for participants to complete the three instruments used in the study. On average, the survey took approximately 15–20 minutes to complete. This information has also been added to the manuscript, as suggested.
Finally, it appears that many data points were collected but not used in your analysis, such as sociodemografic and health data. Please clarify the rationale for collecting these data and specify whether they were intended for future analyses, exploratory purposes, or were later deemed irrelevant. Sociodemographic and health information were initially gathered to provide a broader context for the sample. However, during the analysis phase, it became evident that these variables were not relevant to the specific aims and scope of the present study. As such, they were not included in the final analysis. We have clarified this point in the manuscript to avoid any confusion regarding the rationale for data collection.
Results: The data presented on line 134 (38.7% self-reported chronic diseases), as well as the information from lines 135–139, do not appear in any of the tables. At the same time, sociodemographic data are included in Table 1, which reflects a disorganized presentation of the results. We acknowledge the inconsistency in the initial presentation of the results and apologize for not including the data from lines 134–139 in the corresponding table. We have now incorporated this information into Table 1 to ensure that all relevant data are presented in a more organized and consistent manner.
Furthermore, both the sociodemographic variables and the self-reported general health data seem to have no relevance to your study, as they are not used or analyzed in any meaningful way. I strongly recommend reorganizing the results section — either by grouping participants into age categories or by conducting correlational analyses to identify potentially vulnerable subgroups in terms of communication with the dentist or dental student. Otherwise, it remains merely a descriptive presentation of unused data, which does not contribute meaningfully to the study's objectives or conclusions. You need to be consistent in how you report p-values — for example, line 151 reports p = .007, while line 156 uses p = 0.007. Please choose one format and apply it uniformly throughout the manuscript. We would like to note that other reviewers have expressed different perspectives regarding the use of sociodemographic and general health data. Our study’s primary aim was to assess whether there are differences in communication satisfaction of geriatric patients treated at a university dental clinic when interacting with dentists compared to fifth-year undergraduate dental students. Given this specific focus, we did not conduct subgroup analyses based on age categories or correlate these variables with communication satisfaction, as these analyses would represent a distinct research question and scope beyond the current study. We appreciate the reviewer’s recommendation, and we agree that such analyses could be valuable for future research projects specifically designed to explore vulnerabilities within subgroups. Regarding the reporting of p-values, thank you for pointing out the inconsistency. We have standardized the format throughout the manuscript to ensure uniformity.
Discussion: On line 116, reference number [10] appears to be incorrectly formatted or inconsistent with the citation style used throughout the manuscript — please revise accordingly. You are encouraged to add recent comparative studies and to expand the discussion through relevant comparisons with similar findings in the current literature. We have revised it to ensure consistency with the citation style used throughout the manuscript. We did not find any new studies comparing this type of sample that we could include in our study. However, we also used some ideas from a theoretical study (Ho et al., 2025). We appreciate your helpful suggestions.
Moreover, the limitations of the study are missing and should be clearly stated. Lastly, please provide more insights into how dental academic programs can implement active teaching methodologies that effectively integrate both verbal and non-verbal communication strategies into the curriculum. Thank you for your valuable comments. We agree that stating the study’s limitations is important, and we have now included a dedicated section addressing them in the manuscript. We have also expanded the discussion to provide more insights on how dental academic programs can incorporate active teaching methodologies that effectively integrate both verbal and non-verbal communication strategies into the curriculum.
Conclusions: Please move limitations to the end of the Discussion section, where the study they more appropriately belong (e.g., “research conducted at a single dental university – limited external validity”). Thank you for the suggestion. We have moved the study limitations to the end of the Discussion section, as recommended.
References: I kindly ask you to expand the reference list by including more recent and relevant sources. Additionally, the entire bibliography must be reformatted to match the citation style required by the target journal, as it currently does not conform to the submission guidelines. Thank you for your helpful recommendation. We have added some recent and relevant references, although we believe the original selection already adequately supports our study. Additionally, we thoroughly reformatted the entire bibliography to fully comply with the citation style required by the target journal. We appreciate your guidance in improving the manuscript.
|
|
|
|

Round 2
Reviewer 2 Report
Comments and Suggestions for Authors
Dear authors, thanks to provide a revisited version of your manuscript and to take into consideration my suggestions. I'm satisfied with these.
Author Response
There were no additional comments in the second round. Thank you for your time and support during the review process.
Reviewer 3 Report
Comments and Suggestions for Authors
Dear Authors
all my concerns have been addressed. Now the manuscript ha been improved, written fluently and suitable for publication.
Best regards
Author Response

(The authors gave the same response as above.)

Reviewer 4 Report
Comments and Suggestions for Authors
Please see the attachment

Author Response
Thank you for your valuable suggestion. We have carefully revised the manuscript to improve the language, grammar, and clarity throughout. We believe these changes have enhanced the overall readability of the paper. All edits are highlighted in the revised version.